# A Comprehensive Evaluation of Biomedical Entity Linking Models

**David Kartchner**[1,2*] **Jennifer Deng**[2] **Shubham Lohiya**[2] **Tejasri Kopparthi**[2]

**Prasanth Bathala**[2] **Daniel Domingo-Fernández**[†1] **Cassie S. Mitchell**[† 2]

[1]Enveda Biosciences
[2]Georgia Institute of Technology
david.kartchner@gatech.edu
dani@envedabio.com
cassie.mitchell@bme.gatech.edu

## Abstract

Biomedical entity linking (BioEL) is the process of connecting entities referenced in documents to entries in biomedical databases such as the Unified Medical Language System (UMLS) or Medical Subject Headings (MeSH). The study objective was to comprehensively evaluate nine recent state-of-the-art biomedical entity linking models under a unified framework. We compare these models along axes of (1) accuracy, (2) speed, (3) ease of use, (4) generalization, and (5) adaptability to new ontologies and datasets. We additionally quantify the impact of various preprocessing choices such as abbreviation detection. Systematic evaluation reveals several notable gaps in current methods. In particular, current methods struggle to correctly link genes and proteins and often have difficulty effectively incorporating context into linking decisions. To expedite future development and baseline testing, we release our unified evaluation framework and all included models on GitHub at https://github.com/davidkartchner/biomedical-entity-linking.

## 1 Introduction

Biomedical entity linking (BioEL) is the process of identifying biomedical concepts (e.g. diseases, chemicals, cell types, etc.) in text and connecting them to a unique identifier in a knowledge base (KB). Entity linking (EL) is critical in text mining, as it allows concepts to be connected across disparate literature. This "harmonization" enables quick access to connected information in the knowledge base and allows for unified reasoning regarding diverse surface forms and mentions.

† These authors contributed equally to this work.
* Correspondence: david.kartchner@gatech.edu

While entity linking is a critical task for text mining, BioEL remains an unsolved problem with diverse challenges. First, biomedical literature has complex, specialized jargon that may differ between biomedical subspecialties. This leads to large, varied sets of synonyms that can be used to reference the same entity. For example, the entity ncbigene:37970 can be referred to by the aliases "ORC", "ORC4", "origin recognition complex subunit 4", "CG2917", "rDmORC", "dmOrc4", etc. Moreover, the entity referenced by a particular surface form is context-dependent and may require specialized domain expertise to disambiguate. For instance, within the Unified Medical Language System (UMLS), "AD" could refer to *Alzheimer's Disease*, *Atopic Dermatitis*, *Actinomycin D*, or *Admitting Diagnosis*.

Second, annotating a biomedical corpus is a time-consuming task that requires specialized domain expertise, which have limited availability to label data. Concretely, the largest labeled BioEL dataset, MedMentions (Mohan and Li, 2019), covers approximately 1% of the candidate entities in its reference ontology while annotating 0.17% of the abstracts in PubMed.

Third, though dozens of ontologies and terminologies have been curated in recent years, concepts are often not cross-referenced, leading to a lack of interoperability. Furthermore, even carefully unified collections such as UMLS lack synonyms and definitions for the vast majority of concepts.

Most biomedical concepts are not labeled in any gold-standard EL corpus. Thus, robust zero-shot performance is critical for effectively performing EL at scale. However, lack of labelled data by specialized domain experts simultaneously makes it

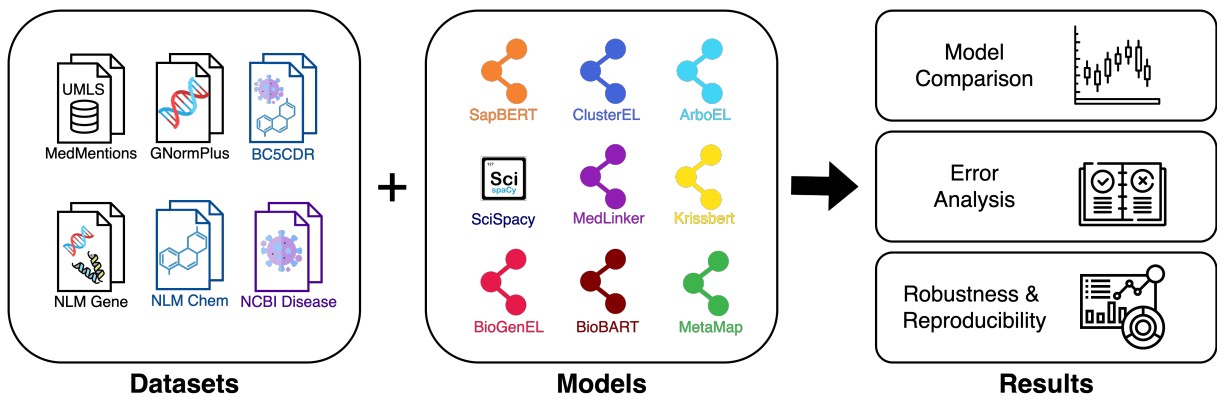

Figure 1: Overview of BioEL evaluation framework.

difficult to accurately assess the capacity of current models to generalize to unseen data.

While some BioEL surveys have been published (French and McInnes, 2022), they do not evaluate models in a consistent way or on a uniform collection of datasets. Rather than a traditional survey, we contend a systematic *evaluation* of current BioEL models is needed to: 1) accurately compare current models; 2) identify strengths and weaknesses; 3) prioritize directions for future research; 4) provide a framework to expedite future BioEL development. To address these needs, this paper contributes the following:

- We release a synthesized collection of current BioEL models, which can be uniformly evaluated on a large collection of biomedical datasets.

- We present a systematic framework to evaluate entity linking models along axes of *scalability*, *adaptability*, and *zero-shot robustness* (Section 5).

- We conduct, to our knowledge, the largest and most comprehensive comparative evaluation of BioEL models to date.

- We highlight strengths and pitfalls of current BioEL modeling techniques and suggest directions for future improvement (Section 7).

- We provide our unified framework as open source repo to expedite future BioEL method development and baseline testing.

## 2 Problem Definition

We assume that we are given a corpus $\mathcal{D} = \{d_i\}_{i=1}^N$ of text, where each $d_i$ is a document in the cor-

| Symbol | Definition |
|--------|------------|
| $\mathcal{D}$ | Corpus of documents |
| $d_i$ | Individual document in corpus |
| $m_{ij}$ | An entity mention in document $i$ |
| $c_{ij}^{-(+)}$ | Left (right) context of entity mention $m_{ij}$ |
| $\mathcal{M}$ | Collection of all entities in context |
| $\mathcal{E}$ | Database of entities |
| $e_k$ | Individual entity |

Table 1: Notation used throughout paper

pus (e.g. a clinical note, biomedical research abstract, etc.). Each document is annotated with mentions spans $m_{ij} \in d_i$, where every mention span $m_{ij} = t_{ij}^{(1)}, \ldots, t_{ij}^{(\ell)}$ is a sequence of tokens corresponding to a single entity. Every mention is given with surrounding contextual information $c_{ij}^-$ and $c_{ij}^+$, which correspond to token spans before and after the entity mention $m_{ij}$. Define the collection of contextual mentions for a document $M_i = \{c_{ij}^- m_{ij} c_{ij}^+\}_{j=1}^{n_j}$. Subsequently, we discuss mentions within the context of a single document and thus drop the document subscript $i$ from mention and context annotations.

We assume that a database of entities is provided $\mathcal{E} = \{e_k\}_{k=1}^K$. Each entity is identified by a unique identifier and may also contain informational metadata such as entity type(s), definition, aliases, etc. Most entity-linkers assume access to ground truth entity mention spans. However, these can be determined programmatically via a named entity recognition algorithm.

The task of entity linking is to learn a function $f : \mathcal{M} \to \mathcal{E}$ that maps each mention $m_j$ to the correct entity $e_j \in \mathcal{E}$.

Most entity linkers use a two-stage approach to find the correct entity link for a given mention span.

The first stage is **Candidate Generation** (CG), which defines a function $f_{CG} : \mathcal{M} \to \mathcal{E}^n$ that filters $\mathcal{E}$ down to a set of $n$ high-quality candidate entities. Once a set of entity candidates have been generated, they are passed into a **Named Entity Disambiguation** [*] (NED) module $f_{NED} : \mathcal{E}^n \times \mathcal{M} \to \mathcal{E}$, which chooses the best candidate for a final entity link. In practice, $f_{CG}$ is chosen to be a computationally inexpensive algorithm with high recall, while $f_{NED}$ is more costly and precise. The final entity linker is defined as $f = f_{NED} \circ f_{CG}$.

## 3 Datasets

We evaluate included BioEL methods on a variety of biomedical datasets (Table 2), with detailed descriptions of each in the Appendix A. All datasets used were taken from BigBio (Fries et al., 2022). Additionally, Table 10 in the appendix describes the extent to which entities and mentions overlap between the training and testing data. Entity overlap is defined as the proportion of entities in the testing data that are in the training data. Mention overlap represents the proportion of *mentions* in the testing data whose *entity* is present in the training data (e.g. if an entity is mentioned more than once in the test set).

### 3.1 Data Preprocessing

In order to simplify data processing, we pulled all included datasets from BigBio (Fries et al., 2022), a recent effort to unify the format of biomedical text datasets for improved consistency and ease of use. Any bug and error fixes for included datasets were contributed directly to BigBio. For KBs, we downloaded the KBs to which each database is linked, namely UMLS (Bodenreider, 2004), MeSH (Lipscomb, 2000), Entrez Gene (Maglott et al., 2005), and the MEDIC dictionary (Davis et al., 2019), which contains disease entities from MeSH and OMIM (Hamosh et al., 2005). The KBs used for each dataset are listed in Table 2.

We removed any entity mentions whose Concept Unique Identifiers (CUIs) were no longer available in the corresponding ontology or remapped them to the updated CUIs when possible. We used Ab3P (Sohn et al., 2008) to identify and (optionally) resolve abbreviations at train/inference time.

In Entrez gene, we additionally dropped "tRNA" and "hypothetical protein" gene types that were not

used for entity linking. For methods able to process additional metadata (ArboEL, ClusterEL), we add species information for each gene in the entity description. For alias matching methods, we added the species name of each gene to its canonical name when the canonical name was not unique. We did not augment other aliases with species information.

### 3.2 Excluded Datasets

This evaluation focuses on entity linking in biomedical scientific research articles (BioEL). Therefore, this systematic evaluation excludes EL in non-scientific texts. Additionally, text extracted from electronic health records (EHR), such as notes or discharge summaries, are also excluded. EL for EHR is distinct from BioEL in its scope, purpose, and accessibility. Previous EHR EL efforts for informal, patient-generated text include CADEC (Karimi et al., 2015), AskAPatient (Limsopatham and Collier, 2016), and PsyTAR (Zolnoori et al., 2019). These EHR EL platforms link diseases, symptoms, and adverse drug reaction mentions to a variety of relevant ontologies. Similarly, COMETA (Basaldella et al., 2020) links a diverse array of entities in Reddit posts to SNOMED-CT.

## 4 Models

A wide variety of methods have been used for BioEL. Here we describe families of models used for BioEL and list included models from each category. More detailed descriptions of each individual model are found in Appendix B. We summarize the different models evaluated in Table 3.

Models evaluated were those with near state-of-the-art performance at time of publication when evaluated on at least one included BioEL entity linking dataset. From this pool, we excluded models with no open-source implementation or whose implementation was rendered unusable due to lack of documentation or software updates. With the exception of MetaMap, all models were published in the past 5 years.

### 4.1 Alias Matching EL

Alias based entity linking seeks to link entities by matching an entity mention with a correct entity alias in a KB. The simplest form of this is exact string matching, but can be extended using any model that produces similarity scores between a mention and a set of candidate aliases. Evaluated alias matching methods include MetaMap (Aron-

---

[*]Note that some works focus only on NED and assume that candidates are given by an existing model.

| Dataset | Num Docs | Mentions | Unique Ents | Ent Types | Doc Type | Ontology |
|---|---|---|---|---|---|---|
| MedMentions Full | 4,392 | 352,496 | 34,724 | 127 | PubMed Abstracts | UMLS |
| MedMentions ST21PV | 4,392 | 203,282 | 25,419 | 21 | PubMed Abstracts | UMLS |
| BC5CDR | 1,500 | 29,044 | 2,348 | 2 | PubMed Abstracts | MeSH |
| GNormPlus | 533 | 6,252 | 1,353 | 2 | PubMed Abstracts | Entrez |
| NCBI Disease | 792 | 6,881 | 789 | 4 | PubMed Abstracts | MEDIC |
| NLM Chem | 150 | 37,999 | 1,787 | 1 | PMC Full-Text | MeSH |
| NLM Gene | 550 | 15,553 | 3,348 | 5 | PMC Full-Text | Entrez |

Table 2: Summary of datasets used for evaluation.

| | Model Characteristics | | Data Requirements | | | Reproducibility Code | | | Usability | |
|---|---|---|---|---|---|---|---|---|---|---|
| Model | Supervised | Type | Names | Definitions | Aliases | Preprocessing | Model Source | Pretrained Model | Documentation | New Dataset |
| MedLinker | Yes | Contextualized | Yes | Yes | Yes | No | Yes | No | Fair | No |
| SciSpacy | Yes | Alias Match | Yes | Optional | Yes | N/A | Yes | Yes | Excellent | Yes |
| ClusterEL | Yes | Contextualized | Yes | Optional | Optional | Yes | Yes | No | Good | No |
| ArboEL | Yes | Contextualized | Yes | Optional | Optional | Yes | Yes | No | Good | No |
| KRISSBERT | Distant | Contextualized | Yes | Optional | Optional | No | Partial | Yes | Good | No |
| BioSyn | Distant | Alias Match | Yes | No | Yes | Yes | Yes | Yes | Good | No |
| SapBERT | Distant | Alias Match | Yes | No | Yes | Yes | Yes | Yes | Good | Partial |
| BioBART | Yes | Autoregressive | Yes | No | Yes | No | Yes | Yes | Poor | No |
| BioGenEL | Yes | Autoregressive | Yes | No | Yes | No | Yes | No | Fair | No |

Table 3: Comparison of model characteristics, reproducibility, and usability

son and Lang, 2010), SciSpacy (Neumann et al., 2019), BioSyn (Sung et al., 2020), and SapBERT (Liu et al., 2021). Note that BioSyn is included via SapBERT since the latter is a higher-performing edition of BioSyn.

## 4.2 Contextualized EL

Much of the work in transformer-based EL has built upon seminal works in zero-shot EL using semantic similarity between contextualized mentions and entity descriptions (Logeswaran et al., 2019; Wu et al., 2020). These methods use entity description metadata to generate and disambiguate entity candidates without the use of alias tables or large-scale supervised mentions, making it easier to generalize EL beyond the scope of training data. Wu et al. (2020) in particular uses a pretrained BERT bi-encoder (Devlin et al., 2019) model to generate candidates by encoding similarity between mentions and descriptions. It then uses a more expensive cross-encoder model to disambiguate candidates for the final entity link. Our evaluation includes MedLinker (Loureiro and Jorge, 2020), ClusterEL (Angell et al., 2021), ArboEL (Agarwal et al., 2022), and KRISSBERT (Zhang et al., 2021). We also note that Bootleg (Varma et al., 2021; Orr et al., 2021) has been used for biomedical entity linking but do not include it due to lack of code for configuring/running their published BioEL models.

## 4.3 Autoregressive EL

First proposed by (Cao et al., 2021), autoregressive EL uses a generative language model to map the text of each mention to its canonical entity name, rather than identifying the index of the correct database entity. It claims the potential to better accommodate additions to a database because an existing model can easily normalize to new entity names without needing to re-train a final output layer. Autoregressive EL can also preform alias matching by training on an alias table potentially reducing the need for hand-labeled training data. Our survey includes BioGenEL (Yuan et al., 2022b) and BioBART (Yuan et al., 2022a).

## 5 Evaluation Strategy

As noted in (Zhang et al., 2021), evaluation strategies between different entity linking papers are inconsistent, leading to wide disparities in reported results. Differences primarily revolve around how to score predictions where multiple normalizations are given for a named entity, e.g. because all predicted entities share the same alias. We identified three main strategies for this in the literature.

1. **Basic** resolves ties by randomly ordering all equally ranked entities.

2. **Relaxed** counts an entity link as correct if *any* of the predicted normalizations match *any* of

the ground-truth normalizations for a given entity.

3. **Strict** counts a normalization as correct only if *all* predicted normalizations match ground-truth normalizations for a given entity. Same as *basic* if no equally ranked normalizations.

For each dataset, we generate ranked entity candidates from each model in Sec. 4. For models that only natively link to UMLS, links to other KBs are computed by predicting entities in UMLS (Bodenreider, 2004) and mapping these predictions to other KBs using cross references provided by UMLS and OBOFoundary (Smith et al., 2007) . Predictions are ranked and evaluated using recall @ k for $k \in \{1, 2, \ldots, 10\}$ (note that recall@1 is equivalent to accuracy). We perform our main evaluations using the *basic* evaluation strategy unless otherwise specified.

## 5.1 Error Analysis

For errors in the dataset, we analyze the following:

**Stage of EL failure:** For incorrectly linked mentions, did the failure occur in CG or NED phase? For failures that occur in candidate generation phase, what proportion of generated candidates have the correct semantic type/semantic group?

**Failure subgroups:** When a model fails, can we identify slices with high/low chances of failure? Inspired by Orr et al. (2021) and Chen et al. (2021), we investigate possible failure modes including:

- *Entity type*. Are entities of particular types frequently linked incorrectly? Are generated candidates in the correct semantic type/group?

- *Popularity*. How often are incorrectly linked entities present in the training data?

- *Available metadata*. Do incorrectly linked surface forms match aliases in the KB? Are KB entities with few aliases and/or no definition more likely to be incorrectly linked?

**Common Misunderstandings:** There are some cases where all models in our comparison find the incorrect entity link in our data. We manually examined cases where all BioEL models provided an incorrect entity link and describe common mistakes made by current BioEL models.

## 6 Results

Our main result in Table 4 shows the recall@1 (accuracy) and recall@5 of each model across all of the datasets. This estimates how well models perform both on candidate ranking and overall candidate generation. Here ArboEL outperforms most models across the majority of datasets. An additional visualization of how recall@k changes for for $k = 1, \ldots, 10$ is shown in Figure 2.

### 6.1 Performance on specific entity types

While most of the datasets evaluated contain only 1-2 entity types, MedMentions contains 127 distinct entity types split into 10 semantic groups. Similarly, both NLM-Gene and GNormPlus link gene mentions from many different species. We compared whether models perform better on specific semantic groups (MedMentions) or on genes from specific species (NLM-Gene). The results are shown in Tables 5 and 12 (Appendix) respectively.

### 6.2 Performance on entities with limited metadata

We analyzed the models' performance on different data slices, as described in section 5.1. Linked entities are biased towards more commonly seen entities, which enables more robust extrapolation of model zero-shot performance and performance on entities with limited metadata (e.g. aliases, definitions, etc). Results for MedMentions ST21PV are shown in Table 6.

## 7 Discussion

Of the models evaluated, there was no model that clearly performed "best" for all datasets or evaluation metrics. However, ArboEL showed consistently high performance and was always among the highest-performing models on each dataset. SapBERT was arguably the best-performing alias matching method, sometimes surpassing ArboEL in recall@5 for various datasets.

One noteworthy result is the relatively poor performance of all models in Table 4 on gene recognition. For alias matching models we see significant increases in recall@k as k increases on both NLM-Gene and GNormPlus than we do for any other datasets. We hypothesize this is due to gene aliases being poorly differentiated between species. This is supported by the steeply increasing recall@k performance of autoregressive and alias-matching

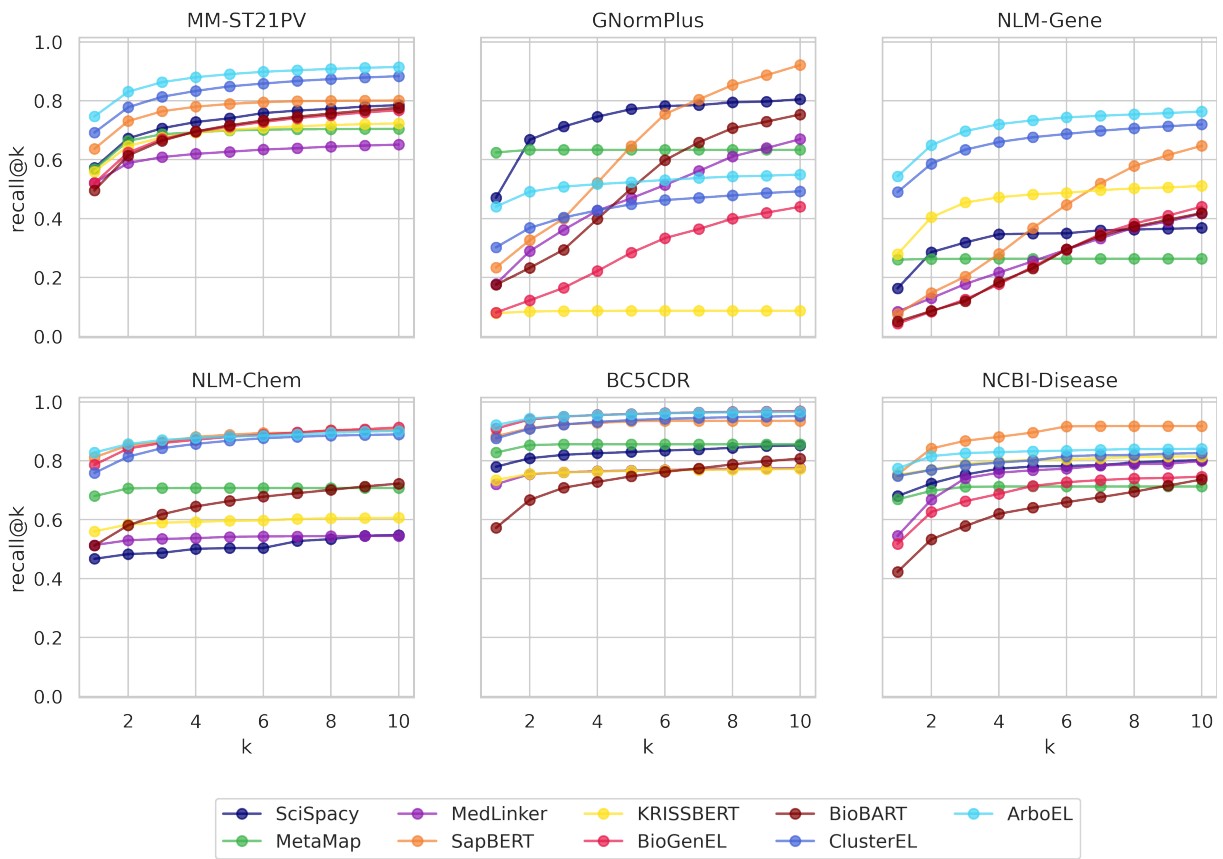

Figure 2: Recall@K for all models using *basic* evaluation.

| | BC5CDR | | MM-Full | | MM-ST21PV | | GNormPlus | | NLM-Chem | | NLM-Gene | | NCBI-Disease | |
|---|---|---|---|---|---|---|---|---|---|---|---|---|---|---|
| | 1 | 5 | 1 | 5 | 1 | 5 | 1 | 5 | 1 | 5 | 1 | 5 | 1 | 5 |
| **SapBERT** | 0.883 | 0.934 | 0.611 | 0.786 | 0.637 | 0.788 | 0.234 | 0.614 | 0.812 | **0.889** | 0.075 | 0.348 | 0.753 | **0.896** |
| **MetaMap** | 0.828 | 0.856 | 0.588 | 0.731 | 0.568 | 0.699 | **0.624** | 0.633 | 0.680 | 0.707 | 0.261 | 0.263 | 0.669 | 0.712 |
| **KRISSBERT** | 0.735 | 0.766 | 0.591 | 0.755 | 0.559 | 0.701 | 0.079 | 0.087 | 0.560 | 0.596 | 0.279 | 0.482 | 0.752 | 0.803 |
| **SciSpacy** | 0.780 | 0.830 | 0.582 | 0.759 | 0.572 | 0.741 | 0.471 | **0.772** | 0.467 | 0.503 | 0.163 | 0.349 | 0.680 | 0.780 |
| **MedLinker** | 0.720 | 0.767 | 0.568 | 0.662 | 0.521 | 0.627 | 0.178 | 0.469 | 0.514 | 0.542 | 0.084 | 0.255 | 0.545 | 0.768 |
| **ClusterEL** | 0.876 | 0.938 | **0.696** | **0.851** | 0.692 | 0.849 | 0.302 | 0.448 | 0.758 | 0.868 | 0.490 | 0.676 | 0.748 | 0.801 |
| **ArboEL** | **0.921** | **0.958** | NR | NR | **0.747** | **0.890** | 0.441 | 0.524 | **0.828** | 0.882 | **0.543** | **0.734** | **0.774** | 0.832 |
| **BioBART** | 0.572 | 0.733 | 0.548 | 0.764 | 0.496 | 0.700 | 0.175 | 0.499 | 0.512 | 0.650 | 0.051 | 0.229 | 0.423 | 0.608 |
| **BioGenEL** | 0.909 | 0.953 | 0.567 | 0.763 | 0.520 | 0.691 | 0.081 | 0.281 | 0.786 | 0.879 | 0.043 | 0.233 | 0.518 | 0.692 |

Table 4: Recall@1 (accuracy) and recall @ 5 of all models. NR=Not reproducible

| Semantic Group | SapBERT | MetaMap | KRISSBERT | SciSpacy | ClusterEL | ArboEL | BioBART | BioGenEL | Prevalence |
|---|---|---|---|---|---|---|---|---|---|
| Disorders | 0.083‡ | 0.065‡ | 0.026‡ | 0.071‡ | 0.038‡ | 0.033‡ | 0.051‡ | 0.073‡ | 0.202 |
| Chemicals & Drugs | -0.027‡ | -0.011 | -0.103‡ | 0.007 | -0.045‡ | -0.034‡ | -0.101‡ | 0.000 | 0.185 |
| Procedures | -0.097‡ | -0.133‡ | 0.018* | -0.127‡ | -0.019† | -0.009 | -0.039‡ | -0.076‡ | 0.165 |
| Living Beings | 0.063‡ | 0.031‡ | 0.045‡ | 0.043‡ | 0.043‡ | 0.047‡ | 0.100‡ | 0.053‡ | 0.099 |
| Physiology | -0.004 | -0.060‡ | 0.046‡ | -0.001 | 0.040‡ | 0.016 | 0.068‡ | 0.024* | 0.095 |
| Concepts & Ideas | -0.011 | 0.049‡ | 0.060‡ | -0.019 | -0.014 | -0.029‡ | 0.038‡ | -0.018 | 0.092 |
| Anatomy | 0.058‡ | 0.125‡ | 0.047‡ | 0.073‡ | 0.035‡ | 0.031‡ | 0.014 | 0.059‡ | 0.082 |
| Genes & Molecular Sequences | -0.144‡ | -0.098‡ | -0.192‡ | -0.14‡ | -0.152‡ | -0.129‡ | -0.153‡ | -0.249‡ | 0.028 |
| Other | -0.030* | 0.027 | -0.039† | 0.008 | -0.039‡ | -0.032† | -0.040† | -0.112‡ | 0.055 |

Table 5: Performance on different semantic groups within MedMentions. Values represent absolute difference in slice accuracy vs. overall accuracy for each model. *p<0.05; †p<0.01; ‡p<0.001 after Bonferroni correction.

| Slice | SapBERT | MetaMap | KRISSBERT | SciSpacy | ClusterEL | ArboEL | BioBART | BioGenEL | Prevalence |
|---|---|---|---|---|---|---|---|---|---|
| is_abbrev | 0.037‡ | 0.080‡ | -0.062‡ | 0.076‡ | -0.023* | 0.003 | -0.038‡ | 0.023* | 0.091 |
| has_alias_match | 0.280‡ | 0.289‡ | 0.114‡ | 0.298‡ | 0.205‡ | 0.194‡ | 0.064‡ | 0.161‡ | 0.157 |
| no_alias_match | -0.052‡ | -0.054‡ | -0.021‡ | -0.055‡ | -0.038‡ | -0.036‡ | -0.012‡ | -0.030‡ | 0.843 |
| wrong_alias_match | -0.259‡ | -0.213‡ | -0.129‡ | -0.175‡ | -0.156‡ | -0.150‡ | -0.156‡ | -0.213‡ | 0.081 |
| train_text_match | 0.094‡ | 0.082‡ | 0.230‡ | 0.077‡ | 0.124‡ | 0.099‡ | 0.094‡ | 0.077‡ | 0.556 |
| train_entity_match | 0.015‡ | 0.023‡ | 0.163‡ | 0.011‡ | 0.058‡ | 0.046‡ | 0.037‡ | 0.017‡ | 0.774 |
| single_alias | -0.075‡ | -0.117‡ | -0.041‡ | -0.148‡ | 0.005 | -0.031‡ | -0.116‡ | -0.133‡ | 0.096 |
| five_alias_or_less | -0.074‡ | -0.085‡ | -0.055‡ | -0.085‡ | -0.04‡ | -0.051‡ | -0.056‡ | -0.079‡ | 0.448 |
| no_definition | -0.101‡ | -0.157‡ | -0.262‡ | -0.126‡ | -0.158‡ | -0.144‡ | -0.152‡ | -0.113‡ | 0.196 |
| zero_shot | -0.051‡ | -0.08‡ | -0.559‡ | -0.038‡ | -0.200‡ | -0.157‡ | -0.128‡ | -0.059‡ | 0.226 |

Table 6: Performance differential of models on various slices of data, micro-averaged over all datasets. Values represent absolute difference in slice accuracy vs. overall accuracy for each model. $^*p<0.05$; $^†p<0.01$; $^‡p<0.001$ after Bonferroni correction.

| Model | BC5CDR | | MM-Full | | MM-ST21PV | | GNormPlus | | NLM-Chem | | NLM-Gene | | NCBI-Disease | |
|---|---|---|---|---|---|---|---|---|---|---|---|---|---|---|
| | CG | NED | CG | NED | CG | NED | CG | NED | CG | NED | CG | NED | CG | NED |
| SapBERT | 0.552 | 0.448 | 0.462 | 0.538 | 0.546 | 0.454 | 0.058 | 0.942 | 0.511 | 0.489 | 0.141 | 0.853 | 0.257 | 0.743 |
| MetaMap | 0.836 | 0.164 | 0.640 | 0.360 | 0.682 | 0.318 | 0.976 | 0.024 | 0.914 | 0.086 | 0.996 | 0.004 | 0.868 | 0.132 |
| KRISSBERT | 0.860 | 0.140 | 0.541 | 0.459 | 0.628 | 0.372 | 0.991 | 0.009 | 0.894 | 0.106 | 0.668 | 0.332 | 0.744 | 0.256 |
| SciSpacy | 0.613 | 0.383 | 0.430 | 0.566 | 0.441 | 0.555 | 0.331 | 0.669 | 0.819 | 0.181 | 0.729 | 0.267 | 0.590 | 0.407 |
| MedLinker | 0.783 | 0.217 | 0.689 | 0.311 | 0.689 | 0.311 | 0.323 | 0.677 | 0.919 | 0.081 | 0.499 | 0.501 | 0.410 | 0.590 |
| ClusterEL | 0.310 | 0.688 | 0.297 | 0.698 | 0.292 | 0.703 | 0.669 | 0.324 | 0.399 | 0.599 | 0.475 | 0.519 | 0.620 | 0.380 |
| ArboEL | 0.403 | 0.597 | NR | NR | 0.275 | 0.722 | 0.780 | 0.219 | 0.536 | 0.464 | 0.477 | 0.521 | 0.677 | 0.323 |
| BioBART | 0.291 | 0.709 | 0.306 | 0.691 | 0.325 | 0.672 | 0.202 | 0.795 | 0.320 | 0.680 | 0.375 | 0.619 | 0.242 | 0.747 |
| BioGenEL | 0.308 | 0.692 | 0.353 | 0.644 | 0.417 | 0.582 | 0.510 | 0.481 | 0.324 | 0.676 | 0.358 | 0.639 | 0.449 | 0.544 |

Table 7: Stage of model (CG or NED) at which entity linking failed. Values represent the proportion of errors that occurred in each stage. NR=Not reproducible

models, which cannot differentiate between multiple entities containing the same alias. Comparison to the recall@k curves under a relaxed evaluation (Figure 11, Appendix) reveals that these models are excellent at finding the correct alias but lack the capacity to choose the correct entity from among them.

For datasets focusing on chemicals and diseases (BC5CDR, NCBI-Disease, NLM-Chem), curves comparing recall from 1 - 10 flatten out quickly; this result indicates that when the correct candidate is retrieved, it is generally ranked highly.

## 7.1 Failure Stage

Most entity linking models consist of two stages, CG and NED. Therefore, it is useful to see at which stage each model failed. If a model is not choosing a set of candidates with the correct entity in the CG stage, the NED stage will never be able to choose the correct one. Table 7 shows how errors are split between candidate generation and reranking for each model.

Failure stage varies widely by dataset and model. MetaMap and KRISSBERT tend to struggle most with candidate generation while BioBART and BioGenEL make most of their errors in entity disam-

biguation. Other models tend to have more evenly distributed errors, with failure stage being highly dataset dependent. Overall, these results indicate that substantial gains can be made to EL through work on both CG and NED.

## 7.2 Impact of Abbreviation Resolution

Abbreviation resolution (AR) is commonly used as a means to potentially improve the performance of EL models. We investigated to what extent this is true by running each of the models with and without AR. The results, shown in Table 8, indicate that AR has a positive, statistically significant effect overall on EL performance: AR improved performance by up to 69.5% on abbreviated entities in some datasets. However, this was not the case for gene normalization where AR showed a negative or insignificant effect. We hypothesize this is because genes are more commonly referred to by their abbreviations than by their longer full names, which limits the usefulness of AR.

## 7.3 Robustness on Slices + Zero Shot

In addition to AR, we evaluated how models performed on different subsets of the data. Some common entity characteristics, along with their perfor-

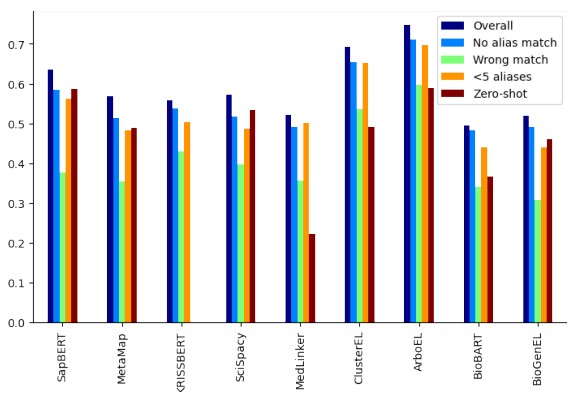

Figure 3: Performance on zero-shot, few alias, and unmatched/mismatched test set instances, evaluated on MedMentions ST21PV.

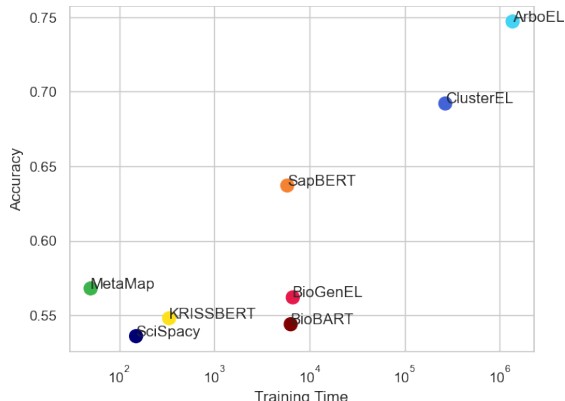

Figure 4: Comparison of training time (s) vs. top-1 entity linking accuracy for BioEL models. All experiments were performed on a single NVIDIA A40 GPU.

mance, are shown in Table 6. A plot of performance in low-data slices (no/wrong alias match in training data; few aliases in KB; zero-shot performance) for MedMentions are shown in Figure 3. Unsurprisingly, we see that the models have significantly improved performance on entities that match an alias in the target ontology; are in the training set; or have definitions. The models performed worse when the mention matches the alias of a different entity; when the ground-truth entity does not have a definition; and when only few aliases are present for an entity in the ontology. We also see that performance degrades in zero-shot settings, but this degradation proportion seems lowest in alias matching models. Overall zero-shot performance is highest on ArboEL, followed by SapBERT.

Taken as a whole, these results indicate that "in the wild" entity linking performance will suffer for entities outside of the training distribution, but these effects can be mitigated by model choice.

### 7.4 Scalability

Scalability is critical for deploying models in practice. To measure the scalability of the models, we compared training and evaluation time on MedMentions. We compared training time in Figure 4 and evaluation time in Figure 5 (Appendix). When a model came pretrained, we include the loading and/or dictionary embedding time as part of its training time. We generally found that simpler alias matching models tended to be faster than autoregressive and contextualized models.

### 7.5 Usability, Adaptability, Reproducibility

We compared the usability and reproducibility of models in Table 3. At the time of our evaluation,

most available research models for EL lacked some or all important elements of reproducibility. For example, a surprising number of models lacked instructions on how to test their method on a different dataset and many models had poor/outdated usage documentation. Some were missing critical details needed to reproduce reported experiments or to simply to run the baseline model. At the time of our evaluation, SciSpacy had the best documentation and use instructions. MedLinker, BioGenEL, and ArboEL were the most difficult to adapt and reproduce.

## 8 Future Work and Conclusion

### 8.1 Future Directions

Large language models (LLMs), such as GPT 3.5 (Ouyang et al., 2022), PaLM (Chowdhery et al., 2022), and BLOOM (Scao et al., 2022) have shown powerful few and zero-shot performance at a variety of tasks. However, these models are known to hallucinate and produce factually incorrect information. To our knowledge, little work has been done to analyze how well these models can correctly link entities, especially biomedical entities that may not be well represented within their training distributions. An evaluation of LLM-based EL stands to improve the performance of BioEL models while also improving the quality and accuracy of LLM-generated text.

### 8.2 Conclusion

Entity linking is an essential task for knowledge-intensive natural language processing and is particularly in scientific and biomedical domains. This paper presents a systematic evaluation of BioEL

| Dataset | SapBERT | MetaMap | KrissBERT | SciSpacy | ClusterEL | ArboEL |
|---|---|---|---|---|---|---|
| BC5CDR | 0.598‡ | 0.588‡ | 0.136‡ | 0.695‡ | 0.329‡ | 0.263‡ |
| MM-Full | 0.426‡ | 0.472‡ | 0.142‡ | 0.408‡ | 0.181‡ | N\A |
| MM-ST21PV | 0.398‡ | 0.454‡ | 0.131‡ | 0.403‡ | 0.187‡ | 0.198‡ |
| GNormPlus | 0.039 | 0.004 | 0.019 | -0.169‡ | -0.039 | 0.004 |
| NLM-Chem | 0.644‡ | 0.602‡ | 0.058‡ | 0.548‡ | 0.33‡ | 0.375‡ |
| NLM-Gene | 0.058 | 0.018 | -0.003 | 0.003 | -0.063 | -0.087 |
| NCBI-Dis | 0.139† | 0.468‡ | 0.035 | 0.381‡ | 0.221‡ | 0.091 |
| Overall | 0.447‡ | 0.464‡ | 0.095‡ | 0.426‡ | 0.22‡ | 0.227‡ |

Table 8: Absolute difference in accuracy on for abbreviated entities after abbreviation resolution of abbreviation resolution. *p<0.05; †p<0.01; ‡p<0.001 after Bonferroni correction.

models along axes of performance, scalability, usability, and robustness, enabling more principled, rigorous development and evaluation of future EL work.

## Limitations

One limitation of our paper is a lack of extensive hyperparameter tuning due to computing constraints. While we did perform early stopping on multiple methods to find the optimal amount of model training, we did not perform an exhaustive hyperparameter search for the models listed. For most models, we followed the parameter choices listed by the authors in their respective papers.

In addition to the general, multi-purpose BioEL models included in this work, there are other models designed to address specific entity types (e.g. genes, chemicals). Such models may be better able to deal with nuances of certain data types, such as species selection for gene/protein BioEL datasets. While these models could offer potential improvements on certain datasets and/or data slices, evaluating them is beyond the scope of this work.

KBs evolve over time with new discoveries and additional curation. While we performed significant manual efforts to identify and either update or remove deprecated entity links within the datasets used, additional curation would be required to ensure that every entity identifier properly aligns with the newer KB versions used when the original was unavailable.

Finally, while there could be benefits from performing multi-task entity linking on a combination of multiple datasets, exploring this option and the challenges associated with aligning multiple KBs is beyond the scope of this work.

## 9 Acknowledgements

This research was funded by the National Science Foundation CAREER grant 1944247 to C.M, the National Institute of Health grant U19-AG056169 sub-award to C.M., the Morningside Center for Innovative and Affordable Medicine at Emory University via the Brown Innovation to Market Fund to C.M., and by the Chan Zuckerberg Initiative grant 253558 to C.M.

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

## A  Datasets

### A.1  Additional Dataset Statistics

Table 9 presents key statistcs about our datasets, particularly about the variety of mentions and abbreviations seen in the datasets

### A.2  Dataset Descriptions

Detailed descriptions of datasets included in our paper are as follows. Table 10 describes overlap of entities and mentions between the train and test sets.

**MedMentions (MM) (Mohan and Li, 2019)** is a collection of 4,392 randomly selected PubMed abstracts linked to the Unified Medical Language System (UMLS). Each abstract is comprehensively

| Dataset | Total Mentions | Unique Mentions | Total Abbreviations | Unique Abbreviations |
|---|---|---|---|---|
| BC5CDR | 29,018 | 5,915 | 2,811 | 388 |
| GNormPlus | 6,252 | 2,180 | 991 | 196 |
| MM-Full | 352,312 | 90,842 | 22,399 | 3,906 |
| MM-ST21PV | 203,185 | 65,947 | 18,701 | 3,398 |
| NCBI Disease | 6,881 | 2,136 | 1,611 | 143 |
| NLM Gene | 15,553 | 5,298 | 2,356 | 462 |
| NLM Chem | 37,999 | 4,706 | 8,684 | 372 |

Table 9: Metadata for each dataset

annotated with all terms from UMLS, making Med-Mentions the largest and most comprehensive EL dataset containing span-level annotations. Due to the diversity of UMLS entity types, some categories are not particularly relevant to the majority of biomedical research (e.g. "Professional Group"). Accordingly, MM is most commonly evaluated on the ST21PV subset, which filters candidate entities to come from 18 high-quality ontologies and to fall under 21 semantic type groups.

**Biocreative V CDR (BC5CDR) (Li et al., 2016)** is a subset of 1,500 abstracts with chemical and disease annotations from the Comparative Toxicogenomics Database. Tagged diseases and chemicals are linked to the MeSH ontology.

**GNormPlus (Wei et al., 2015)** is a benchmark of 694 PubMed abstracts annotated with gene mentions linked to the Entrez ontology of genes. It contains the BioCreative II gene mention (BC2BM) task as a subset and an additional set of 151 annotated abstracts.

**NLM Chem Corpus (Islamaj et al., 2021a)** represents the most diverse gold-standard chemical entity linking corpus. Chemical mentions in 150 PMC full-text articles are normalized to MeSH.

**NLM Gene Corpus (Islamaj et al., 2021b)** is a corpus of over 500 full-text articles with gene mentions linked to Entrez gene.

**NCBI Disease Corpus (Doğan et al., 2014)** links disease mentions in PubMed abstracts to the NCBI disease ontology.

## B  Additional details on included models

Here we provide additional details about the algorithms used by included models to supplement section 4.

| Dataset | Ent. Overlap | Ment. Overlap |
|---|---|---|
| MedMentions Full | 0.6199 | 0.8221 |
| MedMentions ST21PV | 0.5755 | 0.7741 |
| BC5CDR | 0.5300 | 0.7733 |
| GNormPlus | 0.0789 | 0.0838 |
| NCBI Disease | 0.6700 | 0.8156 |
| NLM Chem | 0.4747 | 0.6229 |
| NLM Gene | 0.4819 | 0.5408 |

Table 10: Overlap between entities train and test sets. Mention overlap refers to the proportion of mentions in the test set whose entities are in training set mentions.

A wide variety of methods have been used for BioEL. Here we describe families of models used for BioEL and list included models from each category. Models evaluated were those with near state-of-the-art performance at time of publication when evaluated on at least one included BioEL entity linking dataset. From this pool, we excluded models with no open-source implementation or whose implementation was rendered unusable due to lack of documentation or software updates. With the exception of MetaMap, all models were published in the past 5 years. We summarize the different models evaluated in Table 3.

### B.1  Alias Matching EL

**SciSpacy (Neumann et al., 2019)** SciSpacy is a widely used, off-the-shelf library which offers a diversity of pipelines and models for identifying and linking entities in biomedical documents. SciSpacy jointly performs named entity recognition and abbreviation detection for end-to-end EL. EL is performed using TF-IDF matching on character 3-grams of entity mentions.

**MetaMap (Aronson and Lang, 2010)** MetaMap is a tool developed by the National Library of Medicine (NLM), first used in 1994. It uses nat-

ural language processing to map biomedical entities to concepts in the Unified Medical Language System (UMLS) Metathesaurus. Input undergoes syntactic/lexical analysis, where candidate concepts and mappings are generated from phrases found. MetaMap's usage is highly configurable, both in processing and display options. Output can be shown excluding or restricting semantic types, specific vocabularies, concept unique identifiers (CUIs), etc. Its generation of word variants is thorough, and it is domain independent. On the other hand, MetaMap is limited to the English language. Computational speed is relatively slow, especially in the case where complex phrases are present.

**BioSyn (Sung et al., 2020)**    BioSyn performs EL by normalizing each mention surface form to the best alias seen at training time. It does this via a combination of character-level sparse mention features and learned dense vector representations of each mention and entity, which are trained via an alias table such as the UMLS metathesaurus.

**SapBERT (Liu et al., 2021)**    SapBERT (for "self-alignment pretraining BERT") fine-tunes a BioBERT () model to treat each alias of an entity equivalently and to map entity mentions to an alias contained in UMLS. Zhang et al. (2021) point out that SapBERT is unable to distinguish between aliases shared by multiple entities and returns all entities with an alias matching the normalized surface form.

**Contextualized EL**

**MedLinker (Loureiro and Jorge, 2020)**
MedLinker was one of the first EL works evaluated on MedMentions. It combines a BiLSTM model pre-trained on biomedical literature with approximate string matching from UMLS to conduct zero-shot EL (Mohan and Li, 2019).

**ClusterEL (Angell et al., 2021)**    ClusterEL takes a unique approach to EL by treating linking as a supervised clustering problem. ClusterEL begins by creating an similarity graph of mentions within each document, which is then refined via edge removal until each cluster contains a maximum of one entity. This strategy has the dual benefit of jointly modeling EL with co-reference, enabling the NED model to compensate for failures that may occur in the candidate generation phase of EL. Since original implementation of ClusterEL has been merged into ArboEL, we evaluate ClusterEL

as the graph-based reranking of the candidates retrieved by ArboEL's candidate retrieval biencoder (described below).

**ArboEL (Agarwal et al., 2022)**    ArboEL extends the work in ClusterEL by improving the scalability and training regimen of ClusterEL. While ArboEL uses a bi-encoder similar to (Wu et al., 2020), it also incorporates a training scheme based on a mention-mention similarity graph to identify hard negatives, which ultimately lead to better model precision.

**KRISSBERT (Zhang et al., 2021)**    KRISSBERT presents a self-supervised framework for EL using contrastive learning on distantly supervised entity mentions. After distantly labeling a large number of potential entity links with the UMLS metathesaurus, KRISSBERT learns a set of "prototypes" for each entity by training the model to separate mentions of different entities. They show that this can be extended to a supervised setting without additional fine-tuning by simply swapping noisy prototypes for supervised ones, which achieves performance on-par with the best supervised EL models.

## B.2    Autoregressive EL

**BioGenEL and BioBART (Yuan et al., 2022b,a)**
BioGenEL adapts BART (Lewis et al., 2020) to perform entity linking via sequence-to-sequence modeling. It is trained to generate the correct surface form for an entity mention. **BioBART** uses the same procedure to generate text but additionally provides a BART model with a biomedical vocabulary and pre-trained on biomedical text.

## C    Framework

Our evaluation framework seeks to uniformly evaluate biomedical entity linking datasets by using uniform protocols for 1) dataset processing, 2) ontology processing, and 3) evaluation. All packages are implemented in python. We describe each component of our evaluation framework below.

Our framework's dataset module builds on the BigBio framework (`https://huggingface.co/bigbio`) by adding additional preprocessing to prepare entity linking datasets for effective modeling. It provides APIs for stitching passages into whole documents, deduplicating entity mentions, resolving abbreviations, removing deprecated entities, and contextualizing mentions for modeling.

The ontology processing module of our framework enables different biomedical ontologies such as UMLS, Entrez, and others to be standardized to share common attributes. These attributes include database identifier, semantic type(s), canonical name, aliases, alternate IDs, descriptions, and other metadata such as species. Some of these ontologies are very large with elements distributed across multiple files. Accordingly, we provide APIs for extracting relevant subsets, particularly from UMLS.

The evaluation portion of our framework enables straightforward evaluation of multiple entity linking models across multiple metrics. It creates a standardized format for model outputs as well as an evaluation pipeline that can compute different metrics across the various evaluation strategies described in the paper.

## D  Model Evaluation Details

### D.1  MetaMap

A single-line delimited input text file was generated with the unique text mentions from each dataset. The metadata are shown in Table 6. MetaMap's highly customizable nature means that many parameters can be altered to see the impact on model performance. Six parameters were adjusted for each dataset: model year, semantic types, vocabularies, strict or relaxed model, and term processing (Demner-Fushman et al., 2017). Term processing was added with relaxed model runs, as there was no significant difference between strict and relaxed model performance otherwise. For each run, the NLM data version was used, which includes the full UMLS other than a select number of vocabularies (Demner-Fushman et al., 2017). The 2022AA version was used for all datasets except for MedMentions, as those were originally annotated with the 2017AA UMLS. MetaMap does not handle non-ASCII characters, so we pre-processed input through a Java file that replaces/removes non-ASCII characters. A mapping was generated that keeps track of the terms that are altered, so evaluation can be done correctly.

### D.2  Evaluation Strategy for MetaMap

We performed a grid search over multiple different MetaMap settings, including strict vs relaxed model, term processing, and with/without WSD. WSD did not provide significant improvements in model performance and is not included in

the repository; adding the flag to the MetaMap command would suffice to compare the results. For all datasets, using the relaxed model produced the best results. Four methods of evaluation were tested from toggling two options: 1) ranking mappings first, and/or 2) resolving abbreviations. In addition to candidate concepts, MetaMap generates mappings, which are groups of the most promising candidates. A key point of interest when evaluating MetaMap was seeing whether ranking mappings first would improve evaluation metrics over ranking candidates first. Another salient point was examining the impact of expanding abbreviations. For example, the abbreviation for the chemical OCT can be expanded to 22-oxacalcitriol, which may improve MetaMap performance. The abbreviations within the datasets are expanded from mappings for each PMID, and the expanded forms are added to the original text in each dataset. For each method, we selected the configuration of parameters that maximized recall at 1, which varied between ranking mappings first but almost always resolved abbreviations.

## E  Additional Results, Discussion, and Analysis

### E.1  Runtime Comparison

In addition to training time, we also measured the evaluation time of each included model. The results comparing eval time and accuracy are pictured in Figure 5.

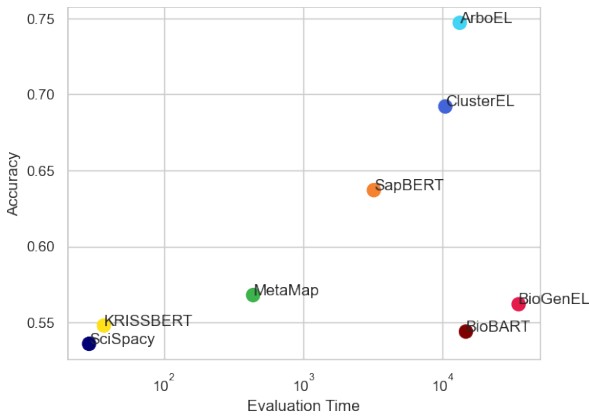

Figure 5: Comparison of evaluation time (s) vs. top-1 entity linking accuracy for

### E.2  Relaxed Evaluation

We provide full results for the models evaluated under a *relaxed* evaluation strategy. A table of

|                | BC5CDR | | MM-Full | | MM-ST21PV | | GNormPlus | | NLM-Chem | | NLM-Gene | | NCBI-Disease | |
|                | 1 | 5 | 1 | 5 | 1 | 5 | 1 | 5 | 1 | 5 | 1 | 5 | 1 | 5 |
|----------------|-------|-------|-------|-------|-------|-------|-------|-------|-------|-------|-------|-------|-------|-------|
| **SapBERT**    | 0.883 | 0.934 | 0.725 | 0.814 | 0.695 | 0.794 | 0.795 | 0.944 | 0.812 | 0.889 | 0.716 | 0.867 | 0.833 | 0.929 |
| **MetaMap**    | 0.828 | 0.856 | 0.588 | 0.731 | 0.568 | 0.699 | 0.624 | 0.633 | 0.680 | 0.707 | 0.261 | 0.263 | 0.669 | 0.712 |
| **KRISSBERT**  | 0.736 | 0.766 | 0.591 | 0.755 | 0.559 | 0.701 | 0.081 | 0.087 | 0.562 | 0.596 | 0.286 | 0.494 | 0.754 | 0.803 |
| **SciSpacy**   | 0.772 | 0.797 | 0.799 | 0.807 | 0.778 | 0.789 | 0.836 | 0.854 | 0.426 | 0.484 | 0.396 | 0.399 | 0.752 | 0.752 |
| **MedLinker**  | 0.720 | 0.767 | 0.568 | 0.662 | 0.521 | 0.627 | 0.178 | 0.469 | 0.514 | 0.542 | 0.084 | 0.255 | 0.545 | 0.768 |
| **ClusterEL**  | 0.876 | 0.938 | 0.696 | 0.851 | 0.692 | 0.849 | 0.302 | 0.448 | 0.758 | 0.868 | 0.490 | 0.676 | 0.748 | 0.823 |
| **ArboEL**     | 0.921 | 0.958 | 0.000 | 0.000 | 0.747 | 0.890 | 0.441 | 0.524 | 0.828 | 0.882 | 0.543 | 0.734 | 0.774 | 0.832 |
| **BioBART**    | 0.572 | 0.733 | 0.662 | 0.800 | 0.544 | 0.711 | 0.696 | 0.847 | 0.512 | 0.650 | 0.521 | 0.714 | 0.457 | 0.689 |
| **BioGenEL**   | 0.909 | 0.953 | 0.686 | 0.793 | 0.562 | 0.698 | 0.350 | 0.527 | 0.786 | 0.879 | 0.504 | 0.698 | 0.582 | 0.733 |

Table 11: Top-1 and top-5 accuracy of all models using relaxed evaluation.

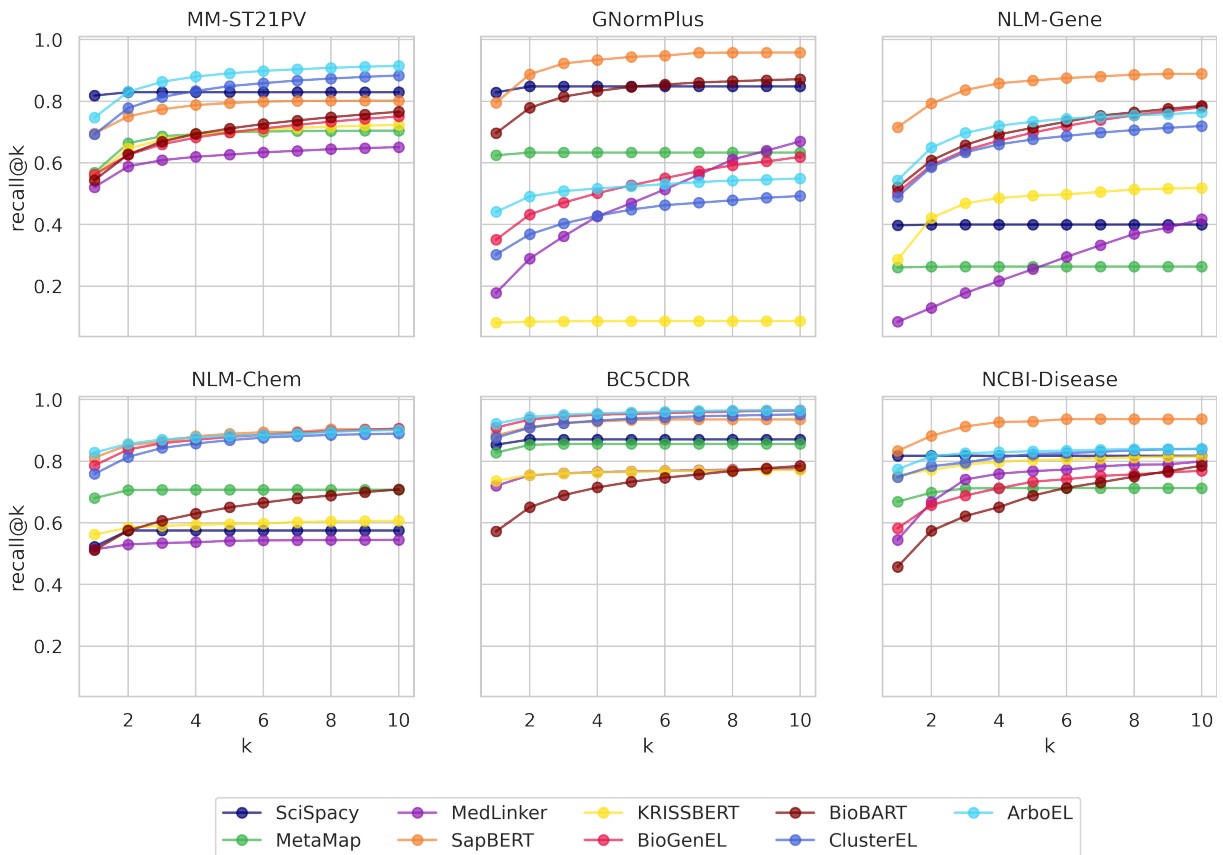

Figure 6: Recall@K for all models using relaxed evaluation.

results is given in Table 11 with a corresponding plot of recall@k in Figure 6.

## E.3 Slice-specific Model Performance

Here we include additional data on the performance of models on various data slices and entity types. Table 12 presents data on performance differentials for different species included in NLM-Gene.

## E.4 Prediction Correlation

It is useful to know to what extent models make similar predictions to know how well they could be ensembled to improve overall results. We accordingly plot the correlation of whether the top-1 predictions match each model. The results, pictured in Figure 7, indicate that models are generally somewhat closely correlated, but differ substantially on gene datasets.

| Taxonomy | SapBERT | MetaMap | KRISSBERT | SciSpacy | ClusterEL | ArboEL | BioBART | BioGenEL | Prevalence |
|---|---|---|---|---|---|---|---|---|---|
| Homo sapiens | -0.021 | 0.307$^{\ddagger}$ | 0.064$^{\ddagger}$ | 0.201$^{\ddagger}$ | 0.125$^{\ddagger}$ | 0.107$^{\ddagger}$ | -0.029$^{\ddagger}$ | -0.014 | 0.447 |
| Mus musculus | -0.048$^{\ddagger}$ | -0.246$^{\ddagger}$ | 0.029 | -0.162$^{\ddagger}$ | -0.010 | 0.016 | -0.040$^{\ddagger}$ | -0.031$^{\ddagger}$ | 0.351 |
| Rattus norvegicus | -0.075$^{\ddagger}$ | -0.244$^{\ddagger}$ | -0.160$^{\ddagger}$ | -0.163$^{\ddagger}$ | -0.249$^{\ddagger}$ | -0.368$^{\ddagger}$ | -0.046$^{\dagger}$ | -0.043$^{\dagger}$ | 0.090 |
| Saccharomyces cerevisiae | 0.046 | -0.261$^{\ddagger}$ | -0.204$^{\ddagger}$ | -0.163$^{\ddagger}$ | -0.256$^{\ddagger}$ | -0.216$^{\ddagger}$ | 0.071$^{\dagger}$ | 0.069$^{\dagger}$ | 0.039 |
| Danio rerio | 0.490$^{\ddagger}$ | -0.261$^{\ddagger}$ | -0.279$^{\ddagger}$ | -0.163$^{\dagger}$ | -0.316$^{\ddagger}$ | -0.225$^{\dagger}$ | 0.573$^{\ddagger}$ | 0.551$^{\ddagger}$ | 0.025 |
| Arabidopsis thaliana | 0.601$^{\ddagger}$ | -0.261$^{\dagger}$ | -0.279$^{\dagger}$ | -0.163 | -0.196 | -0.161 | 0.361$^{\ddagger}$ | 0.045 | 0.012 |
| Ovis aries | -0.038 | -0.261$^{*}$ | -0.279$^{\dagger}$ | -0.163 | -0.045 | 0.086 | -0.014 | -0.006 | 0.010 |
| Caenorhabditis elegans | 0.675$^{\ddagger}$ | -0.261 | 0.021 | -0.163 | -0.190 | 0.157 | 0.549$^{\ddagger}$ | 0.507$^{\ddagger}$ | 0.007 |
| other | 0.365$^{\ddagger}$ | -0.261$^{\ddagger}$ | -0.179$^{*}$ | -0.163$^{*}$ | -0.410$^{\ddagger}$ | -0.323$^{\ddagger}$ | 0.309$^{\ddagger}$ | 0.017 | 0.018 |

Table 12: Performance difference on genes of different species within NLM-Gene compared to overall performance. $^{*}$p<0.05; $^{\dagger}$p<0.01; $^{\ddagger}$p<0.001 after Bonferroni correction.

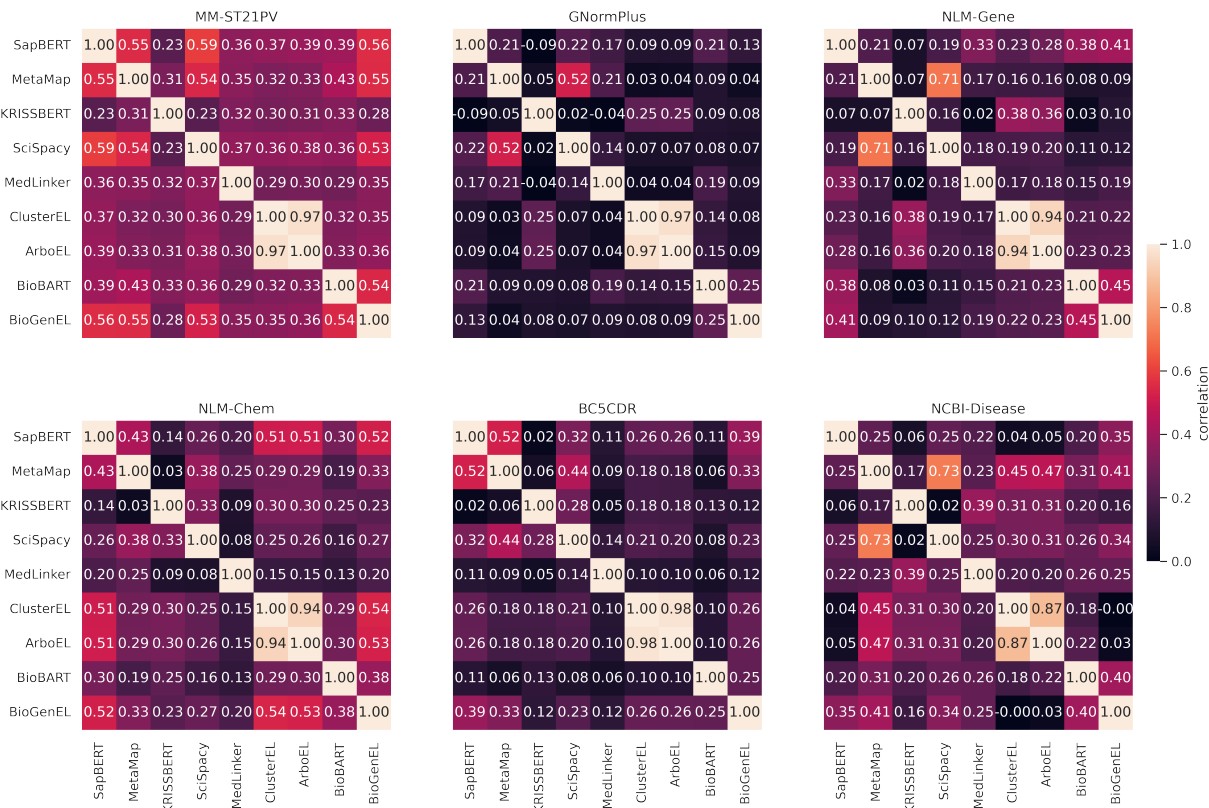

Figure 7: Correlation of top-1 accuracy across datasets. Low and negative correlations indicate that models are able to correctly link distinct subsets of data.