# OpenReview forum: "A Comprehensive Evaluation of Biomedical Entity Linking Models"
_EMNLP/2023/Conference — EMNLP 2023 Main_

### Official Review · Reviewer_wM6u · 2023-07-23

**Soundness:** 3

**Excitement:**

3: Ambivalent: It has merits (e.g., it reports state-of-the-art results, the idea is nice), but there are key weaknesses (e.g., it describes incremental work), and it can significantly benefit from another round of revision. However, I won't object to accepting it if my co-reviewers champion it.

**Missing References:**

1. GNormPlus: An Integrative Approach for Tagging Genes, Gene Families, and Protein Domains

**Paper Topic And Main Contributions:**

- This paper comprehensively evaluates / analyses a set of 8-10 existing methods for Bio entity linking across multiple datasets.

**Questions For The Authors:**

A) What is the previous reported SOTA performance for each of the dataset used in the paper?  For instance, GNormPlus (https://www.hindawi.com/journals/bmri/2015/918710/tab2/) paper reports over 80% Recall, while your paper reports the highest recall@1 as 0.624 in Table 4?

B) How did you choose the models to experiment ? Are they based on previous SOTA performance?

**Reasons To Accept:**

- This is a survey  paper on bio entity linking, the experiments and error analysis are quite thorough.

**Reasons To Reject:**

- The paper claims in the line 16-18, *"current methods struggle to correctly link genes and proteins and often have difficulty effectively incorporating context into linking decisions"*, but the paper does not report the previously published SOTA results (much higher at least for gnormplus dataset https://www.hindawi.com/journals/bmri/2015/918710/tab2/) for each of the dataset. Hence, it is difficult to understand the context of  *"current methods struggle to correctly link genes** . Can you please clarify what you mean by this claim? For instance, is it that a single model / approach does not perform well across all datasets?

**Reproducibility:**

4: Could mostly reproduce the results, but there may be some variation because of sample variance or minor variations in their interpretation of the protocol or method.

**Reviewer Confidence:**

3: Pretty sure, but there's a chance I missed something. Although I have a good feel for this area in general, I did not carefully check the paper's details, e.g., the math, experimental design, or novelty.

---

> ### Author Rebuttal · Authors · 2023-08-27
>
> Thank you for your review and your feedback on our paper.  Below are comments to clarify your questions and concerns.
>
>
> _**The paper claims in the line 16-18, "current methods struggle to correctly link genes and proteins and often have difficulty effectively incorporating context into linking decisions", but the paper does not report the previously published SOTA results (much higher at least for gnormplus dataset https://www.hindawi.com/journals/bmri/2015/918710/tab2/) for each of the dataset. Hence, it is difficult to understand the context of "current methods struggle to correctly link genes" . Can you please clarify what you mean by this claim? For instance, is it that a single model / approach does not perform well across all datasets?**_
>
>
> Thank you for mentioning this.  The table you referenced from the GNormPlus paper reports the performance of _“of **human species** gene normalization”_ **only** as per the figure caption.   Restricting to the subset of human genes is not a fair comparison, as models tend to perform substantially better on this subset (see Table 12 in the Appendix or our paper).  This is not a good representation of the performance of gene entity linking in general.
>
> The authors of the referenced GNormPlus mention this same point: “GNormPlus compares favorably to other state-of-the-art methods when evaluated on two widely used public benchmarking datasets, achieving 86.7% F1-score on the BioCreative II Gene Normalization task dataset and **50.1% F1-score** on the BioCreative III Gene Normalization task dataset”
>
> In light of the above, the referenced claim our the abstract specifically means that current methods struggle to link gene mentions to the correct species.  Species information is generally not given in a gene mention and must be determined by the surrounding context.
>
> We have updated this sentence in the abstract to read "most current methods struggle to link gene mentions to the correct species, often due to failure to incorporate mention context into linking decisions"
>
>
>
> _**A) What is the previous reported SOTA performance for each of the dataset used in the paper? For instance, GNormPlus (https://www.hindawi.com/journals/bmri/2015/918710/tab2/) paper reports over 80% Recall, while your paper reports the highest recall@1 as 0.624 in Table 4?**_
>
> Comparing SOTA across papers is not always straightforward due to differences in evaluation strategies, as described in Section 5 of our paper and in Zhang et al, 2021 (https://arxiv.org/pdf/2112.07887.pdf).  This can be easily seen by examining the difference between Figure 2 and Figure 6 (appendix), which show the same set of results evaluated under different strategies.  Changing the evaluation strategy can result in a performance difference of over 50 percentage points and dramatically shift the ordering of SOTA. This further underscores the importance of this paper to provide a clear comparison of different methods across a uniformly evaluated set of benchmarks.
>
> Thus, one of the key purposes of our paper is to challenge the evaluations presented in previous papers and present a fair comparison of different methods.
>
> As far as we can tell, there are no reported results for all species on the full GNormPlus dataset.  Previous methods have evaluated on the BioCreative II Gene Normalization (BC2GN) dataset, which is a subset of the GNormPlus dataset.  The full GNormPlus dataset includes BC2GN + 151 additional annotated abstracts.  As stated above, the paper you linked only reports recall on human genes.
>
> To clarify this point, we will add a table with the previously reported SOTA + our evaluated performance to the paper, especially highlighting when these differ due to differing evaluation strategies.
>
> _**B) How did you choose the models to experiment ? Are they based on previous SOTA performance?**_
>
> With the exception of MetaMap, chosen were those with near SOTA performance on at least one BioEL dataset at time of publication, published in the past 5 years.  To clarify, we will add the following paragraph to Appendix B:
>
> _“Models evaluated were those with near state-of-the-art performance at time of publication when evaluated on at least one included BioEL entity linking dataset.  From this pool, we excluded models with no open-source implementation or whose implementation was rendered unusable due to lack of documentation or software updates.  With the exception of MetaMap, all models were published in the past 5 years.”_
>
>
> Additionally we chose not to include the GNormPlus and other type-specific models for two reasons:
>
> 1. GNormPlus is only applicable to genes and is not relevant to other biomedical entity linking tasks
> 2. The only version currently available for use is an integrated named entity recognition + entity linking pipeline. This is available via the PubTator API.
>
> For the sake of completeness, we will add a table with the results of GNormPlus and other type-specific models included in the PubTator API.
>
> _**Missing References:
> GNormPlus: An Integrative Approach for Tagging Genes, Gene Families, and Protein Domains**_
>
> The reference you mentioned was already in the paper.  You can find it on lines 718-721.
>
>
> Thank you again for the time taken to review our paper and provide feedback.

---

### Official Review · Reviewer_LFDp · 2023-07-31

**Soundness:** 4

**Excitement:**

4: Strong: This paper deepens the understanding of some phenomenon or lowers the barriers to an existing research direction.

**Paper Topic And Main Contributions:**

I have reviewed this paper in the ACL ARR 2023 April round. The authors have addressed my major comments and added more details in this version.
This paper presents a comprehensive evaluation of the existing models for the biomedical entity linking (BioEL) task. The authors classify nine popular BioEL models into three categories and evaluated them on the seven BioEL datasets. They also conduct an in-depth analysis of the experimental results.

**Reasons To Accept:**

This paper presents a systematic framework to evaluate biomedical entity linking models along axes of scalability, adaptability, and zero-shot robustness. The authors analyze the strengths and pitfalls of these BioEL models and suggest directions for future improvement. The comprehensive evaluation for best practices is valuable in BioEL applications.

**Reasons To Reject:**

The authors have addressed my major comments and added more details in this version. This version is improved.

**Reproducibility:**

4: Could mostly reproduce the results, but there may be some variation because of sample variance or minor variations in their interpretation of the protocol or method.

**Reviewer Confidence:**

5: Positive that my evaluation is correct. I read the paper very carefully and I am very familiar with related work.

---

### Official Review · Reviewer_iWzH · 2023-08-02

**Typos Grammar Style And Presentation Improvements:** Figure 3 could be better presented as…
**Soundness:** 3

**Excitement:**

3: Ambivalent: It has merits (e.g., it reports state-of-the-art results, the idea is nice), but there are key weaknesses (e.g., it describes incremental work), and it can significantly benefit from another round of revision. However, I won't object to accepting it if my co-reviewers champion it.

**Paper Topic And Main Contributions:**

This paper presents a systematic assessment of several entity linking models in the biomedical domain. The work is supported by a "unified evaluation framework", as referred to by the authors, which is not described but which will be made publicly available.
The paper is well organised and the comprehensive evaluation is of interest to the topic.
The evaluation framework will "expedite future BioEL method development and baseline testing"; although no details are given on this framework, this could be a relevant contribution to researcher working on this topic.


**Questions For The Authors:**

It would be important to described how the evaluation framework is structured. Were high level APIs and good practices from popular training/evaluation frameworks followed? Examples that come to mind are Hugging Face and Pykg2vec.


**Reasons To Accept:**

The evaluation framework could be a valuable tool to benchmark new models and/or extend the evaluation.
The paper presents a good summary of the state-of-the-art results in biomedical entity linking.

**Reasons To Reject:**

The framework, which is potentially the most relevant contribution depending on how structured and friendly it is, is not described.
The work is focused on an important but specific topic (within a specific domain), which limits its audience (a biomedical NLP conference could be a more suited forum).

**Reproducibility:**

3: Could reproduce the results with some difficulty. The settings of parameters are underspecified or subjectively determined; the training/evaluation data are not widely available.

**Reviewer Confidence:**

4: Quite sure. I tried to check the important points carefully. It's unlikely, though conceivable, that I missed something that should affect my ratings.

---

> ### Author Rebuttal · Authors · 2023-08-26
>
> Thank you for your helpful feedback on our paper.  We appreciate your recommendation to improve our paper by describing the presented framework.  We have provided an explanation of our framework below, which will be included as an appendix of our paper:
>
>
> *“Our evaluation framework seeks to uniformly evaluate biomedical entity linking datasets by using uniform protocols for 1) dataset processing, 2) ontology processing, and 3) evaluation.   All packages are implemented in python.  We describe each component of our evaluation framework below.*
>
> *Our framework’s dataset module builds on the BigBio framework (https://huggingface.co/bigbio) by adding additional preprocessing to prepare entity linking datasets for effective modeling.  It provides APIs for stitching passages into whole documents, deduplicating entity mentions, resolving abbreviations, removing deprecated entities, and contextualizing mentions for modeling.*
>
> *The ontology processing module of our framework enables different biomedical ontologies such as UMLS, Entrez, and others to be standardized to share common attributes.  These attributes include database identifier, semantic type(s), canonical name, aliases, alternate IDs, descriptions, and other metadata such as species.  Some of these ontologies are very large with elements distributed across multiple files.  Accordingly, we provide APIs for extracting relevant subsets, particularly from UMLS.*
>
>
> *The evaluation portion of our framework enables straightforward evaluation of multiple entity linking models across multiple metrics.  It creates a standardized format for model outputs as well as an evaluation pipeline that can compute different metrics across the various evaluation strategies described in the paper.”*
>
>
> **Additional comments:**
>
> ***The work is focused on an important but specific topic (within a specific domain), which limits its audience (a biomedical NLP conference could be a more suited forum).***
>
>
> While biomedical entity linking is a specialized subdomain of NLP, entity linking is a knowledge-intensive task that is important to the usability and safety of more mainstream NLP technologies, e.g. the correct interpretation and response to medical questions posed to LLMs.  It is our hope that this will enable more entity linking papers to consider biomedical tasks alongside general domain tasks that all build on wikipedia.
>
>
>
> ***Figure 3 could be better presented as a radar chart.***
>
> We believe that a bar plot better represents this data because of sensitivity of radar charts to ordering of variables.  A good example is shown here: https://twitter.com/Koen_VdE/status/1693610007177150721?t=smlerLuqcONEjaEO07NWsg&s=19
>
>
>
> Thank you again for your time to review our paper and provide valuable feedback!

---

### Meta-Review · Area_Chair_uQeE · 2023-09-18

**Recommendation:** 4

**Metareview:**

The paper's comprehensive evaluation of existing Biomedical Entity Linking models is agreed upon by all reviewers, that the framework is a potentially valuable tool benefiting the development of future BioEL methods.

Despite these strengths, the reviewers simultaneously acknowledged certain shortcomings. These include a lack of a precise explanation of the evaluation framework and the audience limitation due to the specific focus of the paper.

In summary, the paper is sound and offers significant insights regarding the BioEL task. However, it is recommended that the authors undertake a revision to address the issues and suggestions raised by the reviewers. Specifically, there is a need for additional clarity regarding some of their claims about current methods' struggles, as well as an accurate reporting of previously published SOTA results for every dataset. Further, a detailed outline of the evaluation framework is necessary.

---

### Decision · Program_Chairs · 2023-10-07

**Decision:**

Accept-Main

**Comment:**

The paper's comprehensive evaluation of existing Biomedical Entity Linking models is agreed upon by all reviewers, that the framework is a potentially valuable tool benefiting the development of future BioEL methods.

Despite these strengths, the reviewers simultaneously acknowledged certain shortcomings. These include a lack of a precise explanation of the evaluation framework and the audience limitation due to the specific focus of the paper.

In summary, the paper is sound and offers significant insights regarding the BioEL task. However, it is recommended that the authors undertake a revision to address the issues and suggestions raised by the reviewers. Specifically, there is a need for additional clarity regarding some of their claims about current methods' struggles, as well as an accurate reporting of previously published SOTA results for every dataset. Further, a detailed outline of the evaluation framework is necessary.